# Genomic Damage Induced in *Nicotiana tabacum* L. Plants by Colloidal Solution with Silver and Gold Nanoparticles

**DOI:** 10.3390/plants10061260

**Published:** 2021-06-21

**Authors:** Petra Lovecká, Anna Macůrková, Kamil Záruba, Tomáš Hubáček, Jakub Siegel, Olga Valentová

**Affiliations:** 1Department of Biochemistry and Microbiology, Faculty of Food and Biochemical Technology, UCT Prague, Technická 5, 166 28 Prague, Czech Republic; anna.macurkova@vscht.cz (A.M.); olga.valentova@vscht.cz (O.V.); 2Department of Analytical Chemistry, Faculty of Chemical Engineering, UCT Prague, Technická 5, 166 28 Prague, Czech Republic; kamil.zaruba@vscht.cz; 3SoWA Research Infrastructure, Biology Centre CAS, Na Sádkách 7, 370 05 České Budějovice, Czech Republic; tomas.hubacek@hbu.cas.cz; 4Department of Solid State Engineering, Faculty of Chemical Technology, UCT Prague, Technická 5, 166 28 Prague, Czech Republic; jakub.siegel@vscht.cz

**Keywords:** comet assay, tobacco, silver nanoparticles, gold nanoparticles

## Abstract

Tobacco seedlings (*Nicotiana tabacum* L cv. *Wisconsin* 38) were treated for 24 h with colloidal solution of silver and gold nanoparticles (AgNPs and AuNPs) of different size or cultivated for 8 weeks on soil polluted with these NPs. DNA damage in leaf and roots nuclei was evaluated by the comet assay. AgNPs of the size 22–25 nm at concentrations higher than 50 mg·L^−1^ significantly increased the tail moments (TM) values in leaf nuclei compared to the negative control. Ag nanoparticles of smaller size 12–15 nm caused a slight increase in tail moment without significant difference from the negative control. The opposite effect of AgNPs was observed on roots. The increasing tail moment was registered for smaller NPs. Similar results were observed for AuNPs at a concentration of 100 mg·L^−1^. DNA damaging effects after growing tobacco plants for 8 weeks in soil polluted with AgNPs and AuNPs of different size and concentrations were observed. While lower concentrations of both types of particles had no effect on the integrity of DNA, concentration of 30 mg·kg^−1^ of AgNPs caused significant DNA damage in leaves of tobacco plants. AuNPs had no effect even at the highest concentration. The content of Ag was determined by ICP–MS in above-ground part of plants (leaves) after 8 weeks of growth in soil with 30 mg·kg^−1^. AgNPs and was 2.720 ± 0.408 µg·g^−1^. Long term effect is much less harmful probably due to the plant restoration capability.

## 1. Introduction

Nanoparticles (NPs) are an important expanding group of new contaminants of the environment. These days, the production and utilization of nanoparticles is increasing in many fields of human activities. This leads to the need to introduce new procedures for assessing their potential risk. The question arises as to whether use to such an extent can be ultimately harmful, as nano-sized chemicals have far different toxic properties compared to their macroscopic counterparts. Studies on their environmental impact are quite frequent. Due to their small size and high reactivity they can have a broad spectrum of impact on the environment and food chain. Compared to particles of larger size it seems to be difficult to predict their toxicity mechanisms. The nanometer size and large surface area facilitate the potential of nanoparticles to interact more effectively with biological systems. Plants have evolved in the presence of naturally occurring nanoparticles [1,2]. With growing production of intentionally produced nanomaterials, however, the probability of their impact on the environment has increased. These fully manufactured nanomaterials can get close to plants by direct application, accidental release into the environment, through contaminated soil or sediments or atmospheric fall [3].

Size-dependent properties of NPs are a widely accepted phenomenon. The size of the smallest nanoparticles (usually below 10 nm, clusters, quantum dots) directly affects the distribution of electron energy levels (quantum constraints) in the material and other phenomena such as semiconductivity, fluorescence, and catalytic activity, while micro-sized materials of the same chemical composition do not exhibit such effects. Silver nanoparticles (AgNPs) are the most commercially available and used as antimicrobial, antiviral, and antifungal agents, for treatment of infections related to burns and open wounds, and for construction of biosensors. AgNPs were studied as agents promoting formation of reactive oxygen species (ROS), the process most often considered to be the cause of metabolic disorders in biological systems. For example, Avalos et al. [4] found the activity of 5 nm AgNPs to be much higher than that of 42 nm ones. Carlson et al. [5] also found that smaller AgNPs (15 nm) are more active in production of ROS than bigger ones, which is in contrast with the work of Oo et al. [6] who studied AuNPs of variable size and obtained opposite results. Plants can be exposed to AgNPs in the environment due to accidental spills, wastewater treatment, agrochemical usage (e.g., nanopesticides, fertilizers, and herbicides), washing, and aging of products containing silver [7].

Manufactured AuNPs have growing application potential in biomedicine, drug delivery, remediation, optical sensing, fuel cell catalysis, and in commercial products. For example, catalytic effects of 20 and 40 nm AuNPs were demonstrated on borohydride reduction of model substrates (Congo red, methylene blue) [8]. Additionally, cellular uptake of AuNPs (18–65 nm in size) by human dermal microvascular endothelial cells [9] and enhanced permeation and retention within permissive tumor microvasculature (24 nm AuNPs and smaller) [10] were proven to be size-dependent. AuNPs exhibit size and concentration dependent cellular activity; cytotoxicity based on both DNA fragmentation and apoptotic activity [11]. Previously mentioned work of Oo et al. [6] demonstrated size-dependent enhancement of ROS formation using 19–106 nm gold spheres modified with protoporphyrin IX. Observed effect of AuNPs on plants is rather positive in respect to plant growth, seed germination rate, seed yield and free radical scavenging activity [12].

Plants interact with soil, water and atmosphere of their environment. NPs present in the atmosphere may form clusters on the surface of the plant, penetrate the plant through the stomata and thus accumulate in the leaves or other above ground parts of the plant [13,14]. NPs are transported via intracellular space and vascular tissue. The distribution of nanoparticles within plants depends on their size and shape [15]. Silver nanoparticles from soil and sediments penetrate the cell wall and plasma membrane of epidermal layers in roots [16]. AgNPs are distributed across the cytoplasm to lysosomes and into the nucleus. They can be genotoxic and potentially induce chromosomal aberrations, micronucleus formation, DNA adduct formation and DNA strand breaks [17].

Nanoparticles, in general, can cause morphological, physiological and genotoxic effects. The tendency of nanoparticles to cross cellular barriers and interact with intracellular structures is given greatly by their small size and high surface reactivity. These also increase their potential to produce cellular and genetic toxicity, most often by induction of oxidative stress [1].

The mechanism of nanotoxicity remains largely unexplored. Probably the most important mechanism is the silver nanoparticle interaction with the cytoplasmic membrane of the cell and with intracellular proteins (membrane sulfur-containing proteins and DNA) [18]. This interferes with cell division and causes cell death. Gold nanoparticles in their conventional forms (e.g., the gold nanoparticles smaller than 20 nm which are red and acquire catalytic properties) show very different characteristics. They enter the cells by endocytosis and in 24 h are enclosed in lysosomes and arranged around the perinuclear region of the cell and are able to induce oxidative damage of DNA as a result of oxidative stress and breaks in DNA chains, depending on nanoparticle size and cell type [19,20].

Many studies have shown that after exposure to NPs various mutations, chromosomal aberrations, DNA breaks, oxidative DNA damage, and DNA adducts occur [21,22]. Metal NPs can react directly with DNA or release metal ions after cell entry, causing genotoxicity. One possibility to test DNA damage is by the Comet assay. It is based on the single-cell gel electrophoresis capable of directly visualizing DNA damage. Nowadays, there are two recognized procedures of this technique: alkaline, developed by Singh et al. in 1988 [23] and the neutral that Olive and Banath [24] introduced in 1990. Alkaline form is used to detect single-strand breaks and neutral for the detection of double-strand breaks. Breaks of DNA molecule occur due to the disruption of the phosphodiester bond between two adjacent deoxynucleotides. Compared to other methods, the Comet test has several advantages such as low cost, ease of execution, low number of cells in the experiment, and high sensitivity in detecting low concentrations of damaged DNA [25].

Our goal was to find out the genotoxic effect of two types of noble metal NPs significantly differing in their reactivity and cytotoxicity with respect to their size. The DNA damage caused by AgNPs and AuNPs of two sizes was performed by Comet assay in tobacco plants cultivated in a hydroponic arrangement and in NPs contaminated soil.

## 2. Results

### 2.1. Characterization of Gold and Silver NPs

Immediately after the synthesis, the AgNPs and AuNPs were studied by transmission electron microscopy (TEM). Figure 1 shows representative TEM images of AgNPs and AuNPs separated from both small (1) and big (2) NP populations. One can clearly see that our synthesis protocol provided quite narrow-size distributed round-shape silver and gold nanoparticles with average sizes ranging from 12 to 15 nm (Ag1, Au1) and 22 to 25 nm (Ag2, Au2) (Figure 1). Nanoparticle average size was determined from TEM images by taking into account at least 1000 particles and excluding size extremes occurring with frequencies below 1%.

A characteristic feature of noble metal nanoparticles is their response to electromagnetic radiation, which is manifested by intense and broad optical absorption bands that arise from the coherent oscillations of conduction electrons near the NP surfaces. This phenomenon is known as surface plasmon resonances (SPRs) and in the case of noble metal NPs the absorption maxima are in the visible region [26]. The shape and position of the SPR band maximum is typical for specific metal NP of defined shape and size [27,28,29]. Therefore, absorption spectroscopy can be advantageously used for characterization of monodisperse, and shape-defined populations of nanoparticles. Typical absorption spectra of prepared silver and gold NPs are shown in Figure 2. The as-synthesized solution of small (Au1) and big (Au2) AuNPs (Figure 2a) exhibited intense absorption bands with the maxima at 525 nm (Au1) and 570 nm (Au2), corresponding well to gold nanoparticles size [28]. Different absorbance values are due to slightly different concentrations of nanoparticles in the analyzed solutions. Pronounced broadening of absorption band of Au2 is due to seedling preparation procedure, reflecting the fact that a negligible amount of starting seeds still contributes to the absorption. The as-synthesized solutions of small (Ag1) and big (Ag2) AgNPs are shown in Figure 2b. Analyzed solutions exhibited intense monomodal peaks with maxima at 403 nm (Ag1) and 416 nm (Ag2), corresponding well to localized plasmon excited in the round shaped silver nanoparticles of corresponding size [27]. Pretty narrow absorption bands point to narrow size distribution of particles in both solutions, which is in good agreement with TEM analysis. Slight red shift of the peak maximum from 403 to 416 nm is due to increase of the particle diameter in solution Ag2.

### 2.2. Short-Term Genotoxic Effect of NPs

#### 2.2.1. AgNPS

Ag particles of two different size ranges (AgNPs 1 of size range 12–15 and AgNPs 2 of size range 22–25 nm) were chosen for testing. After 24 h treatment of tobacco seedlings with concentrations 5, 10, 25, 50, 100 mg·L^−1^ of both NP solutions, nuclei were isolated from roots and leaves, and the Comet assay was performed (Figure 3a). Concentrations of AgNPs 2 (bigger size range) at higher than 50 mg·L^−1^ concentration significantly increased the tail moment (TM) values in leaf nuclei compared to the negative control. The tail moment is defined as the product of the tail length and the fraction of total DNA in the tail (TM = tail length × % of DNA in the tail). AgNPs of smaller size do not show significant differences from control. The opposite effect was observed for nuclei from roots, in which, on the contrary, smaller NPs caused statistically significant increase TM (Figure 3b). For positive control and comparison, H_2_O_2_ was applied on tobacco seedlings for 24 h in concentrations ranging from 0.1 to 0.8 mM. TM values increased significantly in leaf and root nuclei at concentrations above 0.2 mM.

#### 2.2.2. AuNPs

AuNPs of the same size ranges and concentrations as silver ones were tested (AuNPs 1 of size 12–15 nm and AuNPs 2 of size 22–25 nm). After a 24 h treatment of tobacco seedlings with concentrations of 5, 10, 25, 50, 100 mg·L^−1^ of both sizes NPs, nuclei were isolated from roots and leaves, and the comet assay was conducted. We observed the same increasing trends of the dependence of TM on the NPs concentration, (Figure 4a,b). The effect on roots and leaves is less pronounced, no statistically significant differences were found in respect to the size of AuNPs with only exception of the highest concentrations of AuNPs 2. The same effect as for AgNPs was found in roots, where smaller size NP causes an increase in TM in root nuclei comparing to larger AuNPs (Figure 4b).

### 2.3. Long Term Effect of AuNPs and AgNPs on Tobacco Plants

DNA damaging effects after growing tobacco plants for 8 weeks in soil polluted with AgNPs and AuNPs of different sizes and concentrations (7,5, 15, 30 mg·kg^−1^) are illustrated in Figure 5. While lower concentrations of both types of particles had no effect on the integrity of DNA, concentration of 30 mg·kg^−1^ of AgNPs caused significant DNA damage in leaves of tobacco plants. AuNPs had no effect at all concentration.

### 2.4. Content of Au and Ag Ions in Plant Material

The content of Ag and Au ions was determined by ICP–MS in the above-ground part of plants after 8 weeks of growth in soil with NPs content of 30, 15, 7.5 mg·kg^−1^ (Table 1).

## 3. Discussion

NPs size was varied in the range where one can suppose similar genotoxic mechanism based on direct penetration of NPs into intracellular matrix. Therefore, even if we addressed size-dependent effect of Ag and Au nanoparticles, we intentionally kept the NPs size in this interval (<50 nm). It is known that particles larger than 50–80 nm cannot directly penetrate into the cell interior, which inherently distorts obtained data due to a different mechanism of action [18].

It is evident from the literature that several tests for nanotoxicity in plants have been performed. Siddiqi et al. (2016) [12] collected many results by testing effect of nanoparticles on different plant species. Positive results (increase of plant biomass, chlorophyll, protein synthesis and longer root length) as well as negative results were found in some plant species. The observed effects depend on the type, size, shape, and concentration of nanoparticles, but also on the type of plant.

The mechanism of nanotoxicity remains a largely unexplored area. However, it is closely related to the chemical composition, chemical structure, particle size and surface area of the nanoparticles. Toxicity of nanoparticles may be attributed to two different actions: (1) chemical toxicity based on the chemical composition, e.g., release of (toxic) ions; (2) stress or stimuli caused by the surface, size and/or shape of the particles. It has been confirmed that solubility of oxide nanoparticles greatly affects the cell culture response [30].

As already mentioned, NPs can damage DNA either by direct interaction with DNA or through the formation of reactive oxygen species (ROS) and the induction of oxidative stress. Genotoxic potential has also been demonstrated for mammalian cells in various genotoxicity tests for many synthetic nanomaterials [19,31], but in plants such studies are much less frequent. The results of these studies are hardly comparable as they differ in plant species and nanoparticles used. Plants are cultivated in hydroponics or in soil and exposed for different time intervals. NPs themselves differ not only in size, but also in concentrations applied. We compared the genotoxic effect of Ag and Au nanoparticles which were chosen as they differ in their chemical and biological reactivity. Spherical AgNPs and AuNPs of two sizes (12–15 and 22–25 nm) with increasing concentrations were applied to tobacco plants (*Nicotiana tabacum* cv. Wisconsin 38) cultivated in hydroponics in short term experiment or in soil for long term exposure. Genotoxicity was evaluated by the comet assay, which provided evidence that in short term experiment AgNP and AuNP of both sizes significantly induced damage of DNA in concentration dependent manner. Results obtained in this study are in contradiction to those presented by Cvietko et al. [32], who did not observe any significant changes in DNA integrity in tobacco leaves and roots. This is most probably due to their longer time (7 days) of exposure, during which the repair mechanisms could take place.

Interestingly in our experiments the greater effect was observed for smaller particles of both types in roots while in leaves the larger particles caused significant increase in TM compared to smaller NPs. Silver nanoparticles (AgNPs) were shown to induce DNA damages in *A. cepa* and *N. tabacum* with more pronounced defects in roots than in shoots [33]. In leaves the effect of AgNP size was not visible up to the concentration 25 mg/L, the tail moment increase being the same. The increase of TM in leaves was lower than in roots, probably due to the lower content of AgNPs in leaves. This was also reported by [34] for 20 nm AgNPs, which were accumulated in *A. thaliana* leaves 10 times less than in roots. The most pronounced effect was also observed by Ghosh et al. [33] for tobacco roots comparing to leaves for Ag particles of average size of 120 nm at concentrations 50 and 75 mg·L^−1^. However, they did not find any significant difference at concentration 25 mg·L^−1^ which is in a good agreement with our results. Differences in the genotoxicity of AgNPs in root cells compared to *Allium cepa* (onion) and *N. tabacum* (tobacco) shoot cells were also observed using a comet assay in [33]. Changes in the cell structure were also monitored using a transmission electron microscope. Extensive vacuolation, loss of nuclear organization, cracks in the plasma membrane, and protoplast shrinkage were observed in the roots of *A. cepa* plants exposed to silver nanoparticles. The nanoparticles were localized in vacuoles. Sabo-Attwood et al. [35] examined the uptake of AuNPs of different size by the roots of tobacco seedlings by different microscopic techniques. They found that while small particles of 3.5 nm enter the root, particles of 18 nm size accumulate on the surface of the root. The AuNPs of different sizes did not cause any change in DNA damage in leaves but both types of AuNPs caused the increase of tail moment depending on increasing concentration. In roots, the greater effect was observed for larger AuNPs and the effect was comparable that of silver nanoparticles. Very few publications have emerged so far on the effect of Au NPs on plants, to our best knowledge none of them on in vitro DNA damage evaluation. The effect of AuNPs was mostly studied on model system *A. cepa* with respect to MI, chromosome aberrations and oxidative stress. Rajeshwari et al. [36] reported the concentration and size-dependent increase in the chromosomal aberration (CA) frequency in *A. cepa* root tips after treatment with citrate coated AuNPs. Their results, which were obtained under corresponding conditions, for comparable concentrations and AuNPs particle size, are in a good agreement with our findings. Gold nanoparticles in their conventional forms (e.g., when gold nanoparticles are less than 20 nm they are red and obtain catalytic properties) provide a lot of useful information. They enter the cells by endocytosis and in 24 h are enclosed in lysosomes and resting around the perinuclear and are capable of inducing oxidative damage DNA, as a result of oxidative stress and breaks in DNA strands, depending on nanoparticle size and cell type [19,20]. Nevertheless, more studies are needed to draw clear conclusions focused on the genotoxicity of gold nanoparticles.

Long term effect was investigated on tobacco plants cultivated in soil to the stage of four to five true leaves after a single dose of AgNPs at the beginning of the experiment. After 8 weeks the most pronounced effect on DNA damage was observed for smaller AgNPs particles. An increasing trend in tail moment was visible for all types of particles tested, the trend being lowest for AgNPs of bigger size. Significant differences between Ag and AuNPs were found only for the highest concentration used (30 mg·kg^−1^) but for all concentrations tested the effect of AuNPs was lower. This significantly different behavior comparing to the observations in in vitro experiment could be due to the interaction of AuNPs with proteins present in the soil and forming robust coating on the AuNPs surface suppressing the negative effect of NPs [37].

Our experiment also showed (Table 1) that the uptake of AgNPs depends substantially on the NP size even if the differences are relatively small. For the greatest amount of AgNPs applied to the soil, the uptake of smaller particles is more than four times higher. Interestingly, in leaves treated with AuNPs very low amount of Au was determined comparable to those found in plants treated with AgNPs.

In plants, most silver ions accumulate in the root system. This is also reason for the greater sensitivity of plants in the germination phase to the presence of silver [38]. The accumulation of gold by plants is limited, there is very low solubility under natural conditions and very low concentrations in the soil too. The mobility of gold in plant tissues is probably very limited too. AuNPs are chemically inert and do not decompose in the vast majority of environments. Unlike AgNPs, AuNPs do not release ions, thus minimizing any ambiguity between the effects of nanomaterial itself and released ions [39]. AuNPs can be easily functionalized with surface monolayers that prevent direct interaction of AuNPs with the environment and nanoparticle aggregation.

Chemical composition, particle size, shape, surface properties, size distribution, agglomeration state and crystal structure are the most important physical and chemical properties that can influence NPs behavior and their genotoxicity. Recent experiments demonstrated that the NPs concentration, exposition time and NPs size are the most important factors affecting genotoxicity [18,40].

## 4. Materials and Methods

### 4.1. Chemicals and Media

Silver nitrate (puriss; Penta, Prague, CZ), potassium gold(III) chloride (98%; Sigma-Aldrich, St. Louis, MO, USA), sodium chloride (puriss; Penta, Prague, CZ), sodium bromide (puriss; Penta, Prague, CZ), ascorbic acid (99% Sigma-Aldrich, St. Louis, MO, USA), trisodium citrate dihydrate (puriss; Penta, Prague, CZ), normal and low melting-point agarose (Sigma-Aldrich, St. Louis, MO, USA). All chemicals were used as received. All glassware for nanoparticle synthesis was cleaned with *aqua regia* (3:1 *v/v* HCl (3%)/HNO_3_ (65%); Penta, Prague, CZ) and Milli-Q water (18.2 MΩ cm at 25 °C) was used.

### 4.2. Preparation of NPs

*Synthesis of silver nanoparticles **AgNPs1***: Spherical AgNPs were synthesized according to the procedure published by Li et al. [41]. A mixture of citrate (3 mL, 10 mg·mL^−1^), silver nitrate (0.75 mL, 10 mg·mL^−1^), and sodium bromide (3.75 mL, 187 μg·mL^−1^) was added into a boiling solution of ascorbic acid (150 μL of ascorbic acid (17.6 mg·mL^−1^) added to 142.5 mL of boiling water). The transparent and yellow solution was boiled for an hour with stirring, then left to cool down to room temperature.

*Synthesis of silver nanoparticles **AgNPs2***: The procedure was the same as described for ***Ag1*** preparation except the solution of sodium halide. Here, sodium chloride solution instead of sodium bromide solution of the same concentration was used.

*Synthesis of gold nanoparticles **AuNPs1**:* Spherical AuNPs were synthesized according to the procedure published by Bastus et al. [28]. A sodium citrate solution (150 mL, 2.2 mmol·L^−1^) was heated in a 250 mL two-necked round-bottomed flask until it started to reflux. Then, potassium gold(III) chloride solution in water (0.945 mL of 10 mg·mL^−1^) was added. After 30 min, heating was stopped, and the reaction mixture was left to get cold.

*Synthesis of gold nanoparticles **AuNs2:*** Immediately after synthesis of **Au1**, the reaction was left to cool down until the temperature reached 90 °C. Then 1 mL of sodium citrate (60 mM) and 0.945 mL of KAuCl_4_ (10 mg·mL^−1^) were sequentially injected. After 30 min, these additions were repeated, and after additional 30 min they were repeated once more. Then, 55 mL of the reaction mixture was extracted and a citrate solution (55 mL, 2.19 mmol·L^−1^) was added to the remaining reaction mixture. After heating up to 90 °C, potassium gold (III) chloride solution was added. Then, the other two previously described additions of sodium citrate and potassium gold (III) chloride were repeated in 30 min intervals. In the last step, removal of the 55 mL aliquot of reaction mixture and repeated additions of sodium citrate and potassium gold (III) chloride solutions were undertaken once again. Eventually, heating was stopped, and the reaction mixture was left to get cold.

### 4.3. NPs Characterization

After the synthesis, AgNPs and AuNPs were characterized by Atomic Absorption Spectroscopy (AAS), transmission electron microscopy (TEM), ultraviolet–visible spectroscopy (UV–Vis), and mass spectroscopy equipped inductively coupled plasma (ICP–MS).

Concentrations of prepared NPs were determined by means of AAS on a VarianAA880 device (Varian Inc., Palo Alto, CA, USA) using a flame atomizer at 242.8 nm wavelength. Typical uncertainty of concentration determined by this method is less than 3%.

TEM images were taken using JEOL JEM-1010 (JEOL Ltd., Akishima, Japan) operated at 400 kV. Drop of colloidal solution was placed on a copper grid coated with a thin amorphous carbon film on a filter paper. The excess of solvent was removed. Samples were air dried and kept under vacuum in a desiccator before placing them on a specimen holder. Particle size was measured from the TEM micrographs and calculated by taking into account at least 500 particles.

Ultraviolet–visible spectroscopy (UV–Vis) was used to study the optical properties of colloidal dispersions of AgNPs and AuNPs. Absorption spectra were recorded on a Lambda 25 spectrophotometer (PerkinElmer Inc., Waltham, MA, USA) in spectral range 300–850 nm with a 1 nm data step, scan speed of 240 nm/min. Measurements were accomplished in a polystyrene cuvette with 1 cm light path.

Inductively coupled plasma with mass spectroscopy detector (ICP–MS) was used to determine the concentration of Ag and Au ions originating from unreacted source chemical and/or from dissociation equilibrium in nanoparticle colloidal system, using Agilent 8800 triple-quadrupole spectrometer (Agilent Technologies, Santa Clara, CA, USA) connected to an auto-sampler. AgNPs and AuNPs colloidal solutions were pipetted into 3.5 mL microtube, placed into TLA 100.3 rotor and centrifuged at 541,000× *g* on Optima MAX-XP ultracentrifuge (Backman Coulter, IN, USA) for 0.5 h. After this, 1.0 mL of supernatant was carefully removed using a pipette and ICP–MS analyzed. Sample nebulization was performed using a MicroMist device equipped with a peristaltic pump. Pure buffer solution (2.2 mM sodium citrate) was used as a blank sample. The uncertainty of the measurement was less than 5%.

### 4.4. Tobacco Growth and Treatment Conditions

Seeds of the *Nicotiana tabacum* var. Wisconsin 38 plants (were obtained from Institute of Experimental Botany, Prague) were germinated at sterile conditions in plastic vented containers that contained 50 mL of solid Murashige Skoog medium to the stage of 4–5 true leaves.

(1) For 24 h treatments at 22 °C, the roots of tobacco seedlings were immersed in glass vials containing 22 mL of a defined concentration of NPs colloidal suspension. For controls, the seedlings were immersed in distilled water. After treatment roots were 3 times rinsed in water for 5 min and the roots and leaves were used for the comet assay.

(2) For 8 weeks soil experiments, the tobacco seedlings were cultivated in plastic pots filled with 300 g gardening substrate (pH 5–7, humidity 65%, combustible substance in the dried sample 50%). At the onset of the experiment, the soil was watered with 50 mL NPs colloidal suspension of a defined concentration (7.5, 15, 30 mg·kg^−1^). For controls, seedlings were grown in an unpolluted gardening substrate. Six seedlings were cultivated per each concentration of the tested NPs solutions. Plants were cultivated at 22–28 °C in a plant growth chamber with a 16 h photoperiod and were watered with distilled water once a week. For DNA-damage studies, three newly formed leaves were taken from different plants of each treatment variant.

### 4.5. Comet Assay

After treating tobacco seedlings with the tested NPs solutions, excised leaves or roots were placed in a 60 mm Petri dish kept on ice and spread with 250 µL of cold 0.4 M Tris buffer, pH 7.5. Using a fresh razor blade, the leaves or roots were gently sliced, and the isolated nuclei collected in the Tris buffer. The third or fourth true leaves were used for the isolation of nuclei. The preparation of agarose microscope slides with isolated nuclei was earlier described by Gichner et al. [42]. The slides were placed in a horizontal gel electrophoresis tank containing freshly prepared cold electrophoresis buffer (1 mM Na_2_EDTA and 300 mM NaOH, pH > 13). The agarose slide with nuclei were incubated for 15 min to allow the DNA to unwind prior to electrophoresis at 0.72 V cm^−1^ (26 V, 300 mA) for 25 min at 4–8 °C. After electrophoresis, the slides were stained with 80 µL of ethidium bromide (20 µg·mL^−1^) for 5 min, dipped in ice cold water to remove the excess ethidium bromide and covered with a coverslip. For each slide, 25 randomly chosen nuclei were analyzed using a fluorescence microscope with an excitation filter of BP 546/10 nm and a barrier filter of 590 nm. A computerized image analysis system (Comet version 3.1, Kinetic Imaging Ltd., Liverpool, UK) was employed. The tail moment (TM) (integrated value of tail DNA density multiplied by the migration distance) was used as the measure of DNA damage. Three slides were evaluated per treatment and each treatment was repeated at least twice. The TM values are shown as the means of medians ± S.E. Data were analyzed using the statistical functions of SigmaStat 3.0 (SPSS Inc., Chicago, IL, USA). If a significant F-value of *p* < 0.05 was obtained in a one-way analysis of variance test, a Dunnett’s multiple comparison test between the treated and control group was conducted. For all statistical tests, the significance level was established at *p* < 0.05.

### 4.6. The Determination of Ag and Au Ions in Plant Material

An amount of 0.5 g of lyophilized sample of plant material was accurately weighed into a capped 25 mL tube. Concentrated nitric acid (1 mL) was added, and the sample was digested for 30 min at 110 °C. After that, 0.75 mL of diluted HClO_4_ (27 mL of concentrated acid was added to 100 mL of dist. water) and the sample was digested for 2 h at 170 °C. After cooling, the sample was diluted to 3 mL of distilled water and digested for 2 h at 100 °C. The sample solutions were then analyzed using ICP–MS.

## Figures and Tables

**Figure 1 plants-10-01260-f001:**
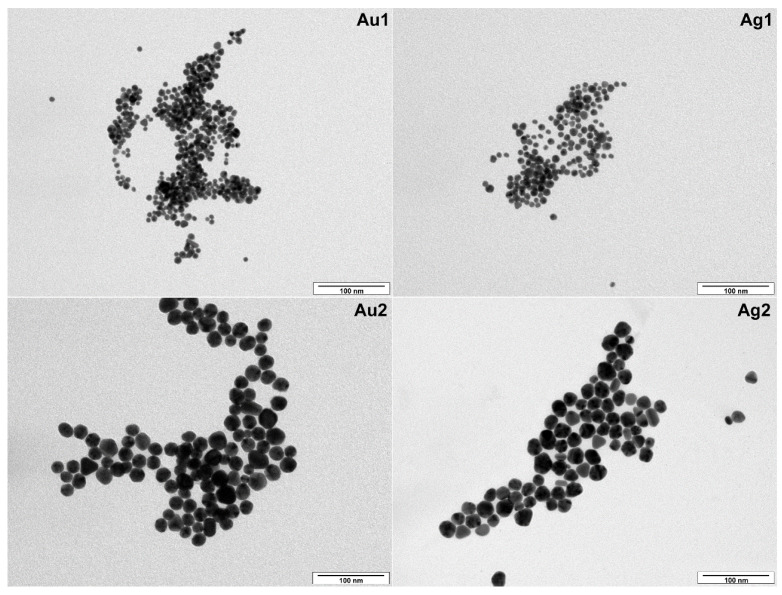
TEM images of AuNPs and AgNPs nanoparticles with average sizes of 12–15 nm (Au1, Ag1) and 22–25 nm (Au2, Ag2).

**Figure 2 plants-10-01260-f002:**
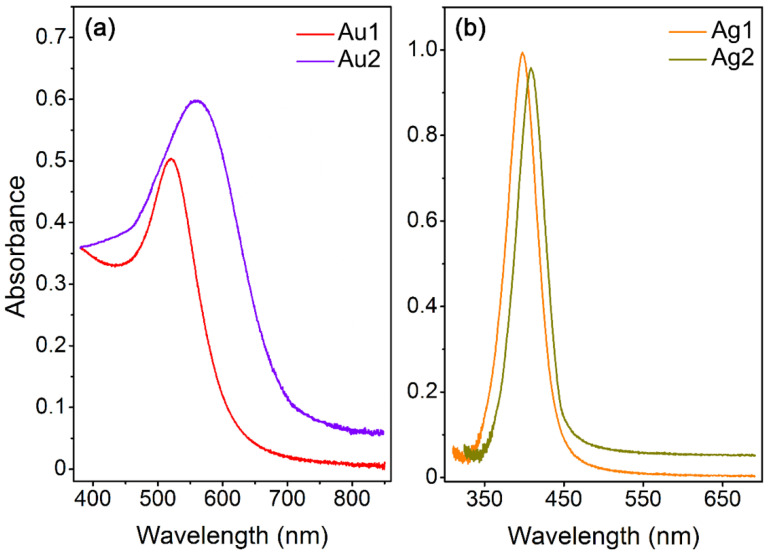
UV–Vis spectra of gold (**a**) and silver (**b**) colloids with average size of AuNPs and AgNPs of 12–15 nm (Au1, Ag1) and 22–25 nm (Au2, Ag2).

**Figure 3 plants-10-01260-f003:**
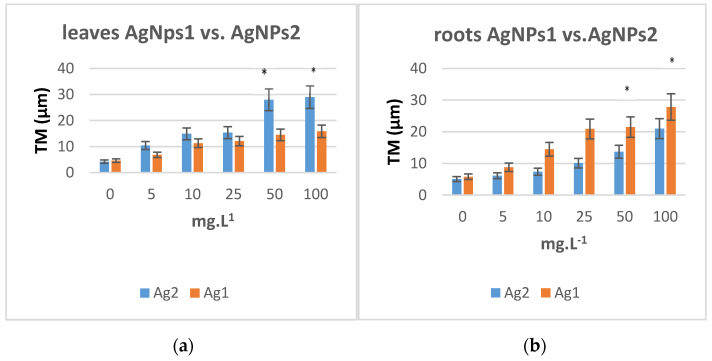
(**a**) The averaged median tail moments (TM) in leaf nuclei after a 24 h treatment at 26 °C of 4–5 weeks old tobacco (*N. tabacum*) plants with aqueous solutions of AgNPs. (**b**) The averaged median tail moments (TM) in root nuclei after a 24 h treatment at 26 °C of 4–5 weeks old tobacco (*N. tabacum*) plants with aqueous solutions of AgNPs (size AgNPs1 12–15 nm and AgNPs2 22–25 nm) The error bars represent the standard error of the mean. (*) Significantly (*p* < 0.05) different from the control.

**Figure 4 plants-10-01260-f004:**
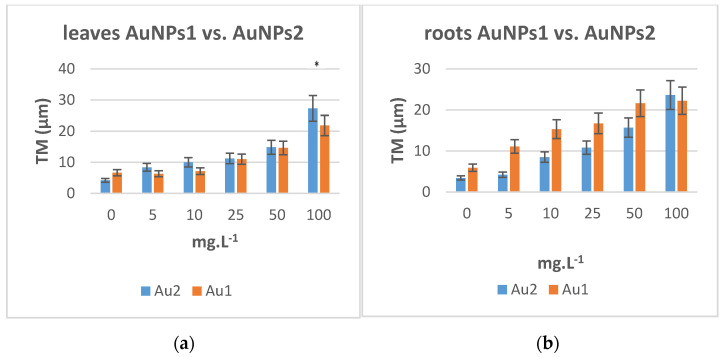
(**a**) The averaged median tail moments (TM) in leaf nuclei after a 24 h treatment at 26 °C of 4–5 weeks old tobacco (*N. tabacum*) plants with aqueous solutions of AuNPs. (**b**) The averaged median tail moments (TM) in root nuclei after a 24 h treatment at 26 °C of 4–5 weeks old tobacco (*N. tabacum*) plants with aqueous solutions of AuNPs (size AuNPs1 12–15 nm and AuNPs2 22–25 nm). The error bars represent the standard error of the mean. (*) Significantly (*p* < 0.05) different from the control.

**Figure 5 plants-10-01260-f005:**
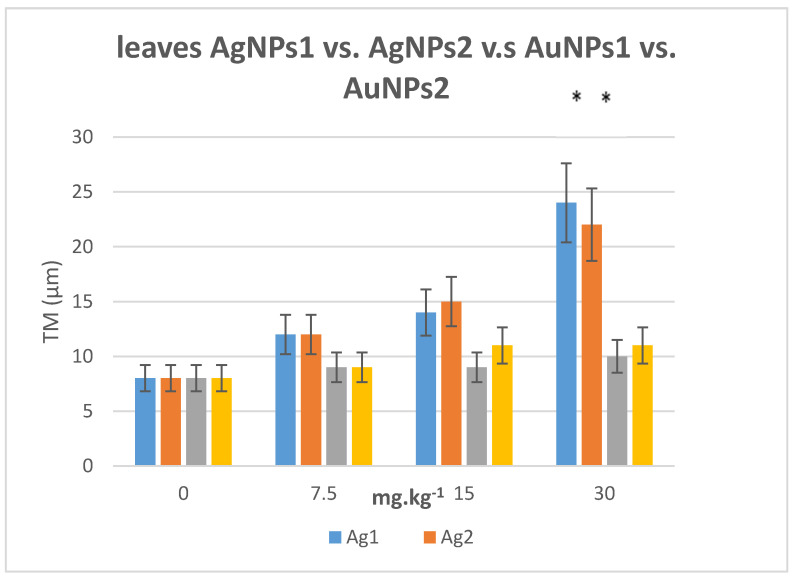
The average median tail moments in leaf nuclei of *N. tabacum* plants cultivated for 8 weeks in soil polluted with aqueous solutions of AgNPs and AuNPs (size AgNPs1 12–15 nm and AgNPs2 22–25 nm, AuNPs1 12–15 nm and AuNPs2 22–25 nm). The plants were cultivated at 22–28 °C in a plant growth room with a 16 h photoperiod. The error bars represent the standard error of the mean. (*) Significantly (*p* < 0.05) different values comparing to the control.

**Table 1 plants-10-01260-t001:** Content of Au and Ag ions (±SD) in leaves after 8 weeks exposure.

Content NPs in Soil [mg·kg^−1^]	Ag1 [µg·g^−1^]		Au1 [µg·g^−1^]
0	0.006 ± 0.001		0.000 ± 0.000
AgNPs1		AuNPs1	
7.5	0.058 ± 0.009	7.5	0.001 ± 0.0001
15	0.280 ± 0.042	15	0.001 ± 0.0001
30	2.720 ± 0.408	30	0.001 ± 0.0001
	**Ag2 [µg·g^−1^]**		**Au2 [µg·g^−1^]**
AgNPs2		AuNPs2	
7.5	0.025 ± 0.004	7.5	0.001 ± 0.0001
15	0.202 ± 0.030	15	0.003 ± 0.001
30	0.798 ± 0.120	30	0.00 1± 0.0001

Size NPs: AgNPs1 12–15 nm, AgNPs2 22–25 nm, AuNPs1 12–15 nm, AuNPs2 22–25 nm, 7.5; 15; 30 … concentration of NPs in soil (mg·kg^−1^).

## Data Availability

The data presented in this study are available on request from the corresponding author.

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
