# Peer review of "Genomic Damage Induced in Nicotiana tabacum L. Plants by Colloidal Solution with Silver and Gold Nanoparticles"

_plants, 2021, doi:10.3390/plants10061260_

Round 1

Reviewer 1 Report

Dear Authors!

I read your manuscript and I marked in yellow my point of view. I ask you to study  the my observations for your final decision.
Best regards.

Author Response

Answers to reviewer's comments:

  1. The concentration of NPs in the experiments is of high importance. How was the concentration for AuNPs and AgNPs determined and are the authors convinced that the studied concentration ratio is sufficiently large? Please discuss in the manuscript

As it is stated in the manuscript, concentration of Au and Ag was determined by Atomic Absorption Sectroscopy.Concentrations of prepared NPs were determined by means of AAS on a VarianAA880 device (Varian Inc., Palo Alto, CA, USA) using a flame atomizer at 242.8 nm wavelength. Typical uncertainty of concentration determined by this method is less than 3%...

The level of NP cytotoxicity is dependent mainly on the following factors: (i) chemical composition, (ii) size, (iii) shape, and (iv) concentration [M. Mahmoudi, K. Azadmanesh, M.A. Shokrgozar, W.S. Journeay, S. Laurent, Effect of nanoparticles on the cell life cycle, Chem. Rev. 111 (2011) 3407–3432.]. The most likely explanation of NP cytotoxic mechanism for living eukaryotic cells has been described for silver NP. In the first step the NP is recognized by surface membrane receptors, then it is incorporated into the plasma membrane, which is followed by translocation into the intracellular space. Inside the cell, NPs accumulate in the organelles, which are subsequently degraded. The most often affected ones are endosomes and lysosomes. The enormous amount of the NPs inside the cell results in a number of adverse effects, such as oxidative stress, cell membrane and DNA damage, cell cycle arrest, apoptosis and/or genotoxicity [N. Lewinski, V. Colvin, R. Drezek, Cytotoxicity of nanoparticles, Small 4 (2008) 26–49.]. Therefore, even if we addressed size-dependent effect of Ag and Au nanoparticles, we intentionally kept the size interval in the range, where NPs may penetrate into the intracellular matrix. It is generally known, that particles larger than 50-80 nm cannot directly penetrate into the cell interior, which inherently distort obtained data due to different mechanism of action [M. Polivkova, T. Hubacek, M. Staszek, V. Svorcik, J. Siegel, Antimicrobial Treatment of Polymeric Medical Devices by Silver Nanomaterials and Related Technology, Int. J. Mol. Sci., 18 (2017).].

  1. The difference observed for Au (inert) and Ag (DNA-toxic) could be rated to a difference in surface chemistry rather than the noble metal material itself. Alternative reasons for genotoxic behavior of colloids should be discussed for the sake of completeness.

Evaluation of the surface chemistry of nanoparticles is a very important issue in the assessment of their genotoxic behavior. Both types of metal nanoparticles (gold and silver) were synthesized in aqueous solutions and both were electrostatically stabilized. Although the two synthetic protocols used were not completely identical, we assume that the only relevant difference was that the surface of the gold nanoparticles was occupied by citrate ions and on the surface of the silver nanoparticles was ascorbate. Since the prepared nanoparticles did not undergo any covalent surface modification and both of these anions were bound only electrostatically, they can be freely exchanged for ions from the surrounding envoronment. From this point of view, we assumed that the metal and its behavior was the main difference and therefore it was reasonable to correlate it with the results of biological experiments.

  1. Today, nanoparticles in biomedical applications are still regarded as a red flag in the eyes of many people. I encourage the authors to discuss the “toxicity” of nanomaterials in a more general context. What do we already know for certain and which new conclusions do their new experiments allow?

Our aim was to investigate the effect on plants, thus we focused the introductory paragraphs on this eukaryotic organisms and did not touch the effect on  animals or humans. Effects on these two different types of eukaryots are hardly comparable at least just because the plant cells are surrounden by quite rigid cell wall. 

  1. How do size effects come into play? The size range studied is very small, at least from the view of nonscientists who usually consider NPs below 40-50 nm as small and everything above as large (up to 100 nm and more). Wouldn’t it make sense to extend the study by such larger colloids (>50 nm) to further improve our understanding of these size effects.

This issue has benn addressed in the reviewer question No. 1

In the view of these facts, we do not consider extension to larger particles (> 50 nm) reasonable. It would certainly bring new interesting results, but this size range was not the subject of our study. This approach would probably require a different design of the whole study

  1. The aspect of particle aggregation owing to colloidal instability in complex biological media/environments is not discussed in enough detail. Tebbe and coworkers (ACS Applied Materials Interfaces) once showed that the protein coating around NPs dramatically modifies their colloidal stability and biotoxocity. I encourage the authors to extend their discussing by including also approaches such as defined protein-coating to control the toxicity of NPs.

A hydroponic variant was used in the short term experiments, roots were after cultivation in MS medium thoroughly washed and immersed in distilled water, so no interaction with proteins is possible - In the long term experiments in soil experiment, aggregation with proteins cannot be excluded.

  1. The “natural” and unavoidably formed protein corona of NPs in contact with biological media could be expected to hinder a contact of DNA to the metal surface. The authors might wish to discuss how much in known (or unknown) about the surface chemistry of their colloids and how it might be changed in the different stages of their experiments.

The answer corresponds with that given to previous comment 5.

  1. Figures 3, 4, 5, and 6 are monochromatic. Color would help. The overall presentation quality is poor. Combining some Figures into large multi-panel figures might help to make the overall flow of the manuscript better and to improve its readability.

We thank to the reviewer for suggestion, figures were corected

  1. I discourage the use of decimal point used in units.

Corrrected in the text.

  1. Section 3 (discussion) is rather lengthy and repetitive. It feels disconnected from the data. Also, the conclusion (section 5) feels too short and superficial. The main findings should be presented and put into context of the state-of-the-art, which is another weak point in the current state.

Corrrected in the text. Conclusion part is not mandatory, we ommit it

  1. Further minor issues: lines 402-404 are repetitions of lines 399-401. L. 377 “.the”; L. 387 typo “mmolL1

           Corrrected in the text

Reviewer 2 Report

Lovecka and coauthors report on the genotoxic effects of noble metals colloids. This was studied in tobacco plant model for gold and silver nanoparticles. The focus of the investigations was on size effects, which are less pronounced, as might be expected. The influence of the surface chemistry was only treated with secondary importance which is a weak point of the manuscript. The biological fate of NPs in plants is complex and the initial surface chemistry as well as the protein corona formed after first contact with the biological medium should be considered. Nevertheless, clear differences in genotoxic behavior between the metals were found. This study fits very well into the scope of the journal plants and I consider a publication here as appropriate. However, the existing uncertainties and open questions should still be clarified before an acceptance can be recommended. My comments below should serve as a basis for an in-depth review.

Comments/questions:

  1. The concentration of NPs in the experiments is of high importance. How was the concentration for AuNPs and AgNPs determined and are the authors convinced that the studied concentration ratio is sufficiently large? Please discuss in the manuscript.
  2. The difference observed for Au (inert) and Ag (DNA-toxic) could be rated to a difference in surface chemistry rather than the noble metal material itself. Alternative reasons for genotoxic behavior of colloids should be discussed for the sake of completeness.
  3. Today, nanoparticles in biomedical applications are still regarded as a red flag in the eyes of many people. I encourage the authors to discuss the “toxicity” of nanomaterials in a more general context. What do we already know for certain and which new conclusions do their new experiments allow?
  4. How do size effects come into play? The size range studied is very small, at least from the view of nonscientists who usually consider NPs below 40-50 nm as small and everything above as large (up to 100 nm and more). Wouldn’t it make sense to extend the study by such larger colloids (>50 nm) to further improve our understanding of these size effects.
  5. The aspect of particle aggregation owing to colloidal instability in complex biological media/environments is not discussed in enough detail. Tebbe and coworkers (ACS Applied Materials Interfaces) once showed that the protein coating around NPs dramatically modifies their colloidal stability and biotoxocity. I encourage the authors to extend their discussing by including also approaches such as defined protein-coating to control the toxicity of NPs.
  6. The “natural” and unavoidably formed protein corona of NPs in contact with biological media could be expected to hinder a contact of DNA to the metal surface. The authors might wish to discuss how much in known (or unknown) about the surface chemistry of their colloids and how it might be changed in the different stages of their experiments.
  7. Figures 3, 4, 5, and 6 are monochromatic. Color would help. The overall presentation quality is poor. Combining some Figures into large multi-panel figures might help to make the overall flow of the manuscript better and to improve its readability.
  8. I discourage the use of decimal point used in units.
  9. Section 3 (discussion) is rather lengthy and repetitive. It feels disconnected from the data. Also, the conclusion (section 5) feels too short and superficial. The main findings should be presented and put into context of the state-of-the-art, which is another weak point in the current state.
  10. Further minor issues: lines 402-404 are repetitions of lines 399-401. L.377 “.the”; L.387 typo “mmolL1”.

Author Response

  1. Chapter 2.3 does not describe the influence of nanoparticles on the average median tail moments in root nuclei

TM was not determined in the roots

  1. Table 1 shows that there are hardly any gold ions in the plant material compared to silver ions. However, Figure 7 shows that there is quite a large influence of Au1 and Au2 on TM. Please explain that.

Incorrect graph processing, corrected.

  1. I do not agree with the sentence: "Au NPs had no effect even with the highest concentration" (page8, line 214). Figure 7 shows something else.

              Incorrect graph processing, corrected.

  1. In chapter 4.6, the beginning of the text is missing. It needs to be completed.

             Text was completed

  1. Chapter 5: Conclusions consists of one sentence. It should be significantly expanded.

As this part is not mandatory, we ommit it

  1. Charts are drawn up carelessly. They need to be improved.

Charts were improved as mentioned above

  1. Some comments about errors:
    • page 3, line 104 – “Many studies have shown” – please list which ones,

two recent review publications were added

following  comments were accepted and text was corrected accordingly

  • page 5, line 165 – there is no explanation for the abbreviation TM, which appears for the first time,
  • page 7, line 205 – “leaf” – it should be "root"
  • page 8, Figure 7 – the plot should be corrected so that the origin of the coordinate system is at the point 0.
  1. Language errors:
    • page 1, title – “tabacco” – it should be "tobacco"
    • page 1, line 37 – “harmfull” – it should be "harmful"
    • page 2, line 67 – “cytalysis” – it should be "catalysis"
    • page 3, line 117 – “nobel” – it should be "noble"
    • page 3, line 124 – “Imediately” – it should be "Immediately"
    • page 3, line 129 – “acount” – it should be "account"
    • page 3,4, line 150, 151 – “which which” – it should be "which"
    • page 4,5 line 159 – “of of” – it should be "of"
    • and many others

correct in text

Reviewer 3 Report

The authors of the article present research on the influence of silver and gold nanoparticles on DNA damage in tobacco plants. The tests were carried out using the Comet assay. The authors' work is interesting and promising because, with the increasing use of nanomaterials, the probability of their impact on the environment increases. Unfortunately, the article is poorly written.

Comments:

  1. Chapter 2.3 does not describe the influence of nanoparticles on the average median tail moments in root nuclei.
  2. Table 1 shows that there are hardly any gold ions in the plant material compared to silver ions. However, Figure 7 shows that there is quite a large influence of Au1 and Au2 on TM. Please explain that.
  3. I do not agree with the sentence: "Au NPs had no effect even with the highest concentration" (page8, line 214). Figure 7 shows something else.
  4. In chapter 4.6, the beginning of the text is missing. It needs to be completed.
  5. Chapter 5: Conclusions consists of one sentence. It should be significantly expanded.
  6. Charts are drawn up carelessly. They need to be improved.
  7. Some comments about errors:
    • page 3, line 104 – “Many studies have shown” – please list which ones,
    • page 5, line 165 – there is no explanation for the abbreviation TM, which appears for the first time,
    • page 7, line 205 – “leaf” – it should be "root"
    • page 8, Figure 7 – the plot should be corrected so that the origin of the coordinate system is at the point 0.
  8. Language errors:
    • page 1, title – “tabacco” – it should be "tobacco"
    • page 1, line 37 – “harmfull” – it should be "harmful"
    • page 2, line 67 – “cytalysis” – it should be "catalysis"
    • page 3, line 117 – “nobel” – it should be "noble"
    • page 3, line 124 – “Imediately” – it should be "Immediately"
    • page 3, line 129 – “acount” – it should be "account"
    • page 3,4, line 150, 151 – “which which” – it should be "which"
    • page 4,5 line 159 – “of of” – it should be "of"
    • and many others

Therefore, I suggest to accept the paper for publication in Plants after following major corrections.

Author Response

Comments were incorporated into the pdf of manuscript and all corrections were accepted  and implemented.

Round 2

Reviewer 1 Report

Dear Authors!

I appreciate the manuscript and I accept  in the present form.

Best regards.

Author Response

thank you for the positive evaluation

Reviewer 2 Report

The authors have provided a revised version, but it is noticeable that most of the changes seem rather cosmetic. On closer inspection this is confirmed and it is also noticeable that the comments have only been answered very superficially. It almost seems as if the authors did not take the reviewer comments and suggestions very seriously. This presumption is based on the high number of typos and the very short and often not very meaningful answers. As a piece of advice, I recommend to change the spell checker in the word processing software from Czech to English. To almost all comments, I would like to reply that I cannot see that the authors have implemented the comment’s suggestions in the manuscript. Even if a comment might be caused by a misunderstanding (of the manuscript), I would strongly recommend that the authors make changes to the manuscript to eliminate the reason for this misunderstanding, otherwise other readers could fall prey to this. In view of the above points, I do not see myself in a position to endorse a publication. In my view, a more detailed revision is necessary.

Comments:

  1. The revised title is grammatically incorrect: „colloidial solutions“.
  2. In reference to comment #1; the authors' reasoning seems reasonable. However, I cannot see any changes to the manuscript that communicate this rationale to the reader. Please revise the manuscript.
  3. In reference to comment #2; the authors' reasoning seems reasonable. However, I cannot see any changes to the manuscript that communicate this rationale to the reader. Please revise the manuscript.
  4. In reference to comment #4; the authors' reasoning seems reasonable. However, I cannot see any changes to the manuscript that communicate this rationale to the reader. Please revise the manuscript.
  5. In reference to comment #5; I am still convinced that aspect of particle aggregation owing to colloidal instability in complex biological media/environments is not discussed in enough detail. Tebbe and coworkers (ACS Applied Materials Interfaces) once showed that the protein coating around NPs dramatically modifies their colloidal stability and biotoxicity. I again encourage the authors to extend their discussing by including also approaches such as defined protein-coating to control the toxicity of NPs.
  6. Comment #6 has not been answered. Please discuss how much in known (or unknown) about the surface chemistry of their colloids and how it might be changed in the different stages of their experiments.
  7. Repeating comment #8, I still discourage the use of decimal points in units. L.214, L.230, L.235, L.284, L.309, Table 1, Fig. 5, and many more instances. Please revise.
  8. Section 3 (discussion) is very lengthy and repetitive. It feels disconnected from the data. The main findings should be presented in a concise collusions section and put into context of the state-of-the-art, which is another weak point in the current state.
  9. The whole manuscript suffers from poor language and plenty of typos and grammatical issues. Please revise.

Author Response

The revised title is grammatically incorrect: „colloidial solutions“.

Corrected accordingly

In reference to comment #1; the authors' reasoning seems reasonable. However, I cannot see any changes to the manuscript that communicate this rationale to the reader. Please revise the manuscript.

In reference to comment #2; the authors' reasoning seems reasonable. However, I cannot see any changes to the manuscript that communicate this rationale to the reader. Please revise the manuscript.

In reference to comment #4; the authors' reasoning seems reasonable. However, I cannot see any changes to the manuscript that communicate this rationale to the reader. Please revise the manuscript.

comments 2-4: previous reasonings were incorporated into the respective parts of the discussion.

In reference to comment #5; I am still convinced that aspect of particle aggregation owing to colloidal instability in complex biological media/environments is not discussed in enough detail. Tebbe and coworkers (ACS Applied Materials Interfaces) once showed that the protein coating around NPs dramatically modifies their colloidal stability and biotoxicity. I again encourage the authors to extend their discussing by including also approaches such as defined protein-coating to control the toxicity of NPs.

As we already mentioned previously, this effect of protein coating of NPs comes into play only for long term experiment in soil, where the proteins occur while the in vitro experiment is performed in distilled water.

We reevaluated the results of long term experiment in the intention of possible protein coating of NP as published by Tebbe et al and reflected it in the discussion.

Comment #6 has not been answered. Please discuss how much in known (or unknown) about the surface chemistry of their colloids and how it might be changed in the different stages of their experiments.

 To our best knowledge, there are no or little information concerning  surface chemistry effect and our ambition was not focused on the study of surface chemistry of colloids .

Repeating comment #8, I still discourage the use of decimal points in units. L.214, L.230, L.235, L.284, L.309, Table 1, Fig. 5, and many more instances. Please revise.

 The text was revised accordingly

Section 3 (discussion) is very lengthy and repetitive. It feels disconnected from the data. The main findings should be presented in a concise collusions section and put into context of the state-of-the-art, which is another weak point in the current state.

The discussion was revised and shortened, to our best knowledge  we discuss our results with published and comparable data. The problem is that the experimental conditions  differ substantially and this fact makes the comparison/discussion difficult.

The whole manuscript suffers from poor language and plenty of typos and grammatical issues. Please revise.

Manuscript was revised by native speaker

Reviewer 3 Report

Thank you for the comprehensive response to comments and questions included in the review. The previous version of the article was re-edited according to reviewers comments and suggestions. Since the authors took remarks included in my review into consideration and added new valuable comments into the text I recommend the presented manuscript for publication in Plants.

Author Response

thank you for the positive evaluation

Round 3

Reviewer 2 Report

The contribution by Lovecka et al has been improved both in terms of content and language. As I am satisfied with the revision, I recommend the acceptance of the manuscript for publication. However, I also suggest some formal changes such as to remove the points in units in the main text as well as in figures/tables. In addition, the references should be cross-checked as some are missing initials or have additional characters (e.g., "Männel, M." and "Chanana, M." in Ref. 37); and the journal name is not correctly abbreviated (ACS Appl. Mater. Interfaces). Ref. 40 is missing the journal name. Ref. 36 seems to be listed before Ref. 35; just to name a few of the many errors.